# Impact of sequential herbicides application on crop productivity, weed and nutrient dynamics in soybean under conservation agriculture in Vertisols of Central India

A. K. Vishwakarma[1], Bharat Prakash Meena [1]*, Hiranmoy Das[2], Pramod Jha[1], A. K. Biswas[1], K. Bharati[3], K. M. Hati[4], R. S. Chaudhary[4], A. O. Shirale[1], B. L. Lakaria[1], Priya P. Gurav[1], Ashok K. Patra[5]

**1** Division of Soil Chemistry & Fertility, ICAR–Indian Institute of Soil Science, Bhopal, India, **2** All India Coordinated Research Project on Soil Test Crop Response, ICAR–Indian Institute of Soil Science, Bhopal, India, **3** Division of Soil Biology, ICAR–Indian Institute of Soil Science, Bhopal, India, **4** Division of Soil Physics, ICAR–Indian Institute of Soil Science, Bhopal, India, **5** ICAR–Indian Institute of Soil Science, Bhopal, India

* bharatmeena24@gmail.com

**Data Availability Statement:** All relevant data are provided within the paper.

## Abstract

Adoption of conservation agriculture (CA) is very slow due to weed infestations. The application of herbicides is the only viable option to deal with problem of weed management to adhere with basic principles of CA. A field experiment was carried out for three years to evaluate the expediency of different herbicides and their sequential applications under CA. In this study, seven treatments comprised of either alone or sequential application of pre-emergence (PE) and post-emergence (PoE) herbicides, hand weeding and weedy check were tested in soybean. Result indicated that sequential application of glyphosate at 1 kg ai ha$^{-1}$ + pendimethalin at 1 kg ai ha$^{-1}$ as PE followed by PoE application of imazethapyr at 100 g ai ha$^{-1}$ at 30 days after sowing (DAS) proved to be the best economical option in terms of plant growth parameters, crop biomass, seed yield, weed index and carbon and nutrient recycling. Pearson's correlation coefficients matrix revealed that grain yield was significantly (P<0.0001) related to weed density at harvest (r = -0.84), (WDH) (r = -0.63), weed dry biomass (WDB) (r = -0.52), weed nitrogen (N), phosphorus (P) and potassium (K) uptake (r = -0.56, r = -0.59 and r = -0.66), respectively and weed index (WI) (r = -0.96). The bivariate linear regression study of grain yield on weed control efficiency (WCI) biomass, N, P and K uptake by grain showed a clear significant (P<0.0001) dependence with R$^2$ value of 0.53, 0.99, 0.95 and 0.98, respectively. The fitted stepwise multiple regression model also revealed that N and P uptake in grain, weed density at 20 DAS and K uptake in weed were actual predictor for grain yield. We concluded that, effective and economical weed control under CA in soybean can be achieved through sequential application of glyphosate along with pendimethalin at 1 kg ai ha$^{-1}$ each PE followed by PoE use of imazethapyr at 100 g ai ha$^{-1}$ at 30 DAS.

**Funding:** The author(s) received no specific funding for this work.

## Introduction

The global population is growing at a faster rate and it has reached to 7.8 billion in 2020. It is estimated to grow further up to 8.5 billion by 2030, 9.7 billion by 2050 and to 10.9 billion by the year 2100 with present trends of population growth rate, [1]. The food production in 2050 will need to increase by 50 percent as compared to 2012 to address the food grain demand. Achieving this projected demand and sustaining the production and productivity levels will be a major challenge in years to come [2]. Adoption of high-yielding varieties, fertilizer application, irrigation and plant protection will remain the most likely options to achieve intensification in agriculture to combat these challenges [3]. However, in recent years the global emphasis has shifted from improving potential yield levels to environmental concerns, soil health, reducing costs of production, and reducing dependency on plant protection measures [4]. Thus, sustainability of future agricultural systems, in the years to come will remain even greater challenge than the present scenario. Tillage has been considered as an essential component under conventional agriculture which makes soil friable, alters weed and nutrient dynamics to help crop establishment and improves physical properties of soil [5,6]. However, the negative impact of tillage and other aspects in conventional agriculture-based production techniques are hidden and are considered as necessary evils [7], However, it adversely affects soil by promoting soil erosion due to its destructive effect of tillage on soil structure and aggregation [8]. Extensive tillage coupled with crop residues burning has resulted in significant reduction in organic matter content and labile carbon pools of the soil [9,10], deterioration in soil physical properties [11] besides being capital- and energy-intensive, tillage accounts for a substantial cost in the total cost of production leading to low economic returns [11]. CA has been considered as one of the most potential technologies which has the potential to not only address the issues related to degradation of natural resource and environmental concerns but will also help in enhancing and sustaining the system productivity [12]. It is aimed at achieving production intensification and high yields together with improvement in the natural resource base by adopting resource-conserving sustainable agricultural production system. Farmers in India can adopt CA looking to multidimensional advantages of the technology through savings in cost of cultivation, improved quality of soil, i.e. physical, chemical and biological parameters [13,14], buildup of soil organic matter and increased carbon (C) sequestration [15–18], reduced infestation of weeds in long term [19] and improved input use efficiency particularly water and nutrients [20,21].

Weed management is considered to be most daunting task under CA compared to conventional agriculture due to the fact that weed seed burial and destruction of vegetative structures by tillage operations are lacking in CA [22]. Compared to conventional tillage (CT); more weed seeds are present on surface of the soil, which resulted in relatively higher accumulation and germination of more weeds under ZT. Hence, CA generally increases weed infestation during initial years due to reduction in tillage intensity [23]. The major hindrance in large scale adoption of CA based production systems is weed management and hence, it is necessary to assess the weed dynamics and their characteristics under CA for successful adoption on large scale [24,25]. Reduction in crop productivity due to increased weed infestation and changes in relative densities of weeds under CA are other major concern restricting spread of CA [26]. Weed control under CA mainly rely on herbicidal weed management [27]. Reduced tillage and greater dependency on herbicides for weed management under CA can bring substantial changes in weed dynamics and their populations [28–30]. Utilization of post-emergence broad-spectrum herbicides in CA based production systems may provide additional opportunity to tackle weed problem [31]. Proper weed management is necessary for obtaining comparable crop yields under conservation tillage system similar to that under conventional tillage system [31]. Thus, proper weed management needs special attention for successful adoption of CA on large scale.

Application of sequential non-selective herbicides for weed control has paved the way for satisfactory crop establishment and consistent yield and creditable performance by providing season-long effective weed control and reducing chances of shift in weed flora towards more prolific weeds due to evolution of herbicide-resistant or tolerant biotypes of weeds [32].

Soybean an important oilseed crops in India cultivated during rainy season (June to October) for its multiple uses and is popular as an oilseed crop. India ranks fifth with 3.95% of global share in production among the soybean producing countries and occupies about 9.0% of the global acreage 120 million hectares, (m ha) under soybean cultivation. India has recorded a production (10.93 million tons) and productivity (1009 kg ha$^{-1}$) of soybean during the year of 2018 [33]. Among various pests, weeds are the most important and vigorous competitors of soybean which germinate and emerge with the crop, established ahead and soon cover the crop which is relatively short stature and susceptible to shading during its initial growth phase [34]. Thus, weeds constitute the most important yield limiting constraint in soybean-based production systems acquiring a major portion of resources [35,36]. In general, weeds are competing for light, nutrients, and moisture during early growth stage with crops and pose early-season competition [37]. Depending upon the types and intensity of weeds crop- weed competition during critical period of crop growth results in 58 to 85%, reduction in yield of soybean and also increase production cost [38]. Losses in soybean due to weed infestation is the major limiting factors responsible for low crop yields, even under conventional agriculture, with all tillage and herbicide usage. Due to rapid urbanization and industrialization laborer's availability at the critical period of crop weed competition (CWC) for weed management is becoming scarce and expensive during peak periods of demand [39,40]. An estimated that the active population engaged in agriculture decreasing day by day, thus restricting availability of economically available workforce in agriculture. Among various pesticides used in agriculture, herbicides account for highest volume eclipsing all other major pesticide groups. Vertisols, being rich in clay (>40%) content are problematic for intercultural operations because of excessive stickiness during rainy season, besides being not permissible in CA. Thus, weed management by applying herbicides is the only viable option for the success of CA in Vertisols. However, among various herbicides there is no herbicide that can provide satisfactory control of different kinds of weeds present in CA field prior to sowing, as well as those which will emerge later and provide season long control [40]. Since weeds cannot be controlled as tillage operations are limited under CA, it is anticipated that the use of herbicides to control weeds under CA will rise [41]. Therefore, it is necessary to standardize effective strategy for control the weeds present before sowing of crop and to take care of weeds that will emerge during the course of crop growth in the presence of crop residue under CA. Under such circumstances, it may not be possible to achieve satisfactory weed control using single herbicide. Thus, it is advocated to use herbicide mixtures to improve the weed control spectrum under CA [42]. For proper weed control and satisfactory crop performance under zero till systems, utilization of proper dose and application of herbicide at right stage is necessary. Since there is lack of scientific information on weed management aspect of CA in Vertisols of central India, experiment was taken up to generate information on effective weed management technology for soybean crop through use of herbicides either alone, their sequential or combined application to improve weed control efficiency, sustaining crop productivity and environmentally safe practice for long term sustainable weed management under CA.

## Materials and methods

### Soil and climate

A field experiment was taken up with soybean crop under CA systems during rainy season 2015 to 2017 at the Research Farm of ICAR-Indian Institute of Soil Science, Bhopal in

Vertisols of Central India. The experimental site is located with latitude of 23˚18´28.26" N, longitude of 77˚24´26.00" E and 485 m above mean sea level. The soil of the experimental site was an *Isohyperthermic*, *Typichaplustert* and deep heavy clay in texture (47.2% clay, 30.5% silt, 22.3% sand), slightly alkaline (pH 8.2) in reaction having a bulk density 1.35 Mg m$^{-3}$, electrical conductivity (EC) 0.18dS m$^{-1}$, soil organic carbon (SOC) 0.68%, available nitrogen 266 kg ha$^{-1}$, phosphorus 27.34 kg ha$^{-1}$ and potassium 524 kg ha$^{-1}$, respectively. The decadal average rainfall in the experimental farm is 1146 mm, more than 80% of which occurs from June to September and potential evapo-transpiration of 1400 mm. The climate in the experimental site is humid subtropical, with warm and humid monsoons during mid of June to end of September.

## Conservation agriculture practices

Sowing was done with the help of 'happy seeder' at the onset of monsoon under no till system. Soybean and wheat crop residue (30%) was maintained uniformly in all the plots irrespective of herbicidal treatments. Variety JS-335 of soybean was planted in the experimental field at a row spacing of 27.5 cm during the last week of June. The recommended dose of chemical fertilizers (30 kg N, 60 kg $P_2O_5$, 40 kg $K_2O$ ha$^{-1}$) has been broadcasted uniformly at the time of sowing. Pre-emergence application of herbicides was done immediately after sowing followed by post-emergence treatment was applied as per scheduled interval. The recommended package of agronomic practices and need based plant protection measures were adopted.

## Treatment details and experimental set-up

The experiment was carried out in a randomized block design comprising of seven treatments and three replications (plot size = 7 m x 7 m = 49 m$^2$). The detailed description of various treatment combinations are presented in Table 1. Observations on plant growth, yield attributes, yield, weed population at different growth stages, crop and weed biomass at harvest were recorded as per standard protocols. Observations on weed density were recorded with the help of a quadrant 0.25m$^2$ placed randomly at four places in each plot. The growth, yield attributes and yields were recorded from net plot area. Plant and seed samples were analysed to determine the nutrient uptake. Weed control efficiency (WCEp) was calculated based on weed population and weed biomass generated at harvest stage using the following formula and

**Table 1. Details description of treatments used in the experiment during study.**

| Treatment | |
|---|---|
| T$_1$ | Absolute control |
| T$_2$ | Two hand weedings at 20 and 40 days after sowing (DAS) |
| T$_3$ | Pre-emergence application of pendimethalin @ 1kg ai ha$^{-1}$ + glyphosate @1 kg ai ha$^{-1}$ |
| T$_4$ | Pre-emergence application of glyphosate @1 kg ai ha$^{-1}$ followed by post- emergence application of propaquizafop @100g + chlorimuron ethyl @ 9 g ai ha$^{-1}$ at 20 DAS |
| T$_5$ | Pre-emergence application of glyphosate @ 1 kg ai ha$^{-1}$ followed by post- emergence application of imazethapyr @100 g ai ha$^{-1}$ at 20 DAS |
| T$_6$ | T$_3$ followed by post- emergence application of imazethapyr @ 100 g ai ha$^{-1}$ at 30 DAS |
| T$_7$ | T$_3$ followed by post emergence application of propaquizafop @ 100 g + chlorimuron ethyl @ 9 g ai ha$^{-1}$ at 30 DAS. |

DAS, days after sowing.

expressed in percentage.

$$\text{WCEp (\%)} = \frac{(\text{WPC} - \text{WPT})}{\text{WPC}} \times 100$$

Where, WCEp: Weed control efficiency population; WPC: weeds population in control plot; and WPT: weed population in treated plot.

$$\text{WCEb (\%)} = \frac{(\text{DWC} - \text{DWT})}{\text{DWC}} \times 100$$

Where, WCEb: Weed control efficiency biomass; DWC: dry weight of weeds in control Plot; and DWT: dry weight of weeds in treated plot.

The weed index (WI) was calculated for different treatments using the formula as suggested by Gill and Kumar [55].

$$\text{WI (\%)} = \frac{(\text{X} - \text{Y})}{\text{X}} \times 100$$

Where, X = Grain yield from weed free check, Y = Grain yield from treatment.

## Collection and analysis of plant samples

Ten random samples of soybean plants and weed samples from quadrats were drawn from each treatment at the time of harvesting, the samples were air-dried, and kept in an oven at 65°C until constant weight was obtained. The sample was powdered with the help of a grinder and these samples were used for the determination of nutrient content. Wet oxidation technique as described by Jackson [43] was adopted for determination of total C content in the samples. Total nitrogen in the crop residues were determined by micro Kjeldhal method after digesting in concentrated sulphuric acid [44]. P and K were determined by digesting the samples in a mixture of $HNO_3$ and $HClO_4$ (9:4) as suggested by Singh *et al.* [45]. The total P was determined by Vanado Molybdate Yellow Colour Method as described by Jackson [46] whereas total K was determined by using flame photometer technique [46] in plant samples. Accordingly, nutrient uptake by soybean was calculated by multiplying nutrient concentration with seed and straw yield. The agronomic use efficiency (AE) and physiological N-use efficiency (PEN) were calculated by the following formulas:

$$\text{AE (kg kg} - 1 \text{ of nutrient applied)} = (\text{GY} - \text{GY0})/\text{FN}$$

$$\text{PE (kg kg} - 1 \text{ of nutrient uptake)} = (\text{GY} - \text{GY0})/(\text{N up} - \text{N0 up})$$

where GY0 and GY represent the seed yield in the N0 plot and fertilized nutrient (NPK) plots, respectively; and FN is the quantity of NPK fertilizer applied in fertilized plot; N0 up and N up are the total nutrient (NPK) uptake in seed and straw in the N0 plot and fertilized nutrient (NPK) plots, respectively.

## Statistical analysis

The 3 years field experiment data on soybean crop and weeds were statistically analysed using SAS 9.3. The square root ($\sqrt{(X + 0.5)}$) transformation was applied on weed density for holding the normality and independence assumptions. Combined/pooled analysis of variance (ANOVA) was carried out through PROC GLM. The treatments mean comparison for different parameters was performed based on Tukey's Honest Significant Difference (HSD) at

P = 0.05 through MEANS statement under PROC GLM. Correlation study was done for understanding the relationship between the parameters by using PROC CORR. Linear bivariate regression analysis was performed between crop yield with WCE and nutrients uptake using PROC REG. The multiple linear regression model was fitted for grain yield by using stepwise selection under PROC REG.

## Results

### Weed flora and weed density

The major weed flora includes monocot weeds like *Echinochloa colonum* (L.) Link., *Dichanthium annulatum* forssk, *Brachiaria plantaginea* Griseb and *Commelina benghalensis* L. and among dicot weeds, *Acalypha indica* L. *Euphorbia geniculata* L., *Phyllanthus niruri* L., *Euphorbia hirta* L., *Digera arvensis Forsk.*, *Alternanthera sessilis* (L.), *Parthenium hysterophorus* L., *Celosia argentea* L., *Caesulia axillaris* Roxb. and *Cyanotis ciliaris* (L.). However, *Cyprus rotundas* (L). was the only weed belonging to sedges category, besides these few minor weeds were also present in the plot under CA. The weed flora was dominated by monocot weeds in the experimental field and recorded 62.3% relative density as against dicot weeds which contribute of the extent of 35% and sedges 2.6%. The most dominant weed among different weed species was *Echinochloa colonum* (L.) Link. with a density of 106.64 m$^{-2}$ sharing 33.70% of total weed population while *Dichanthium annulatum* forssk with a density of 70.64m$^{-2}$ with 22.32% share was second followed by *Acalypha indica* L. (44 plants m$^{-2}$) and relative density of 13.9%. Data pertaining to weed density at different growth stages under various treatments (Table 2) revealed that there was significant effect of different weed management practices under CA. Application of glyphosate alone and in combination with pendimethalin as pre-emergence application resulted in control of already emerged weed in treatments T3, T6 and T7. Weed density at 20 DAS showed that it was on par with respect to T1, T2, T4 and T5 indicating uniformity in weed density among different treatments prior to application of treatment *viz.*, hand weeding and post emergence application of herbicidal weed control treatments. However, treatments T3, T6 and T7 recorded significantly lower values of weed density (5.15, 6.47 and 5.36, respectively) as compared to other treatments at 20 DAS. The significant lower values of weed density in treatments (T3, T6 and T7) indicate reduction in weed density due to pre-emergence application of pendimethalin in these treatments, which were on par with each other. At 60 DAS and at harvest stage all the weed control treatments recorded significantly lower weed densities as compared to T1 (no weeding control) (14.44 and 17.73). Among weed

**Table 2. Effect of different weed management practices on weed densities at different growth stages in soybean.**

| Treatment | Weed density (numbers m$^{-2}$) | | | | | |
|---|---|---|---|---|---|---|
| | **20 DAS** | | **60 DAS** | | **At harvest** | |
| T$_1$ | 12.6$^a$ | (165.1) | 14.4$^a$ | (209.1) | 17.73$^a$ | (316.4) |
| T$_2$ | 12.5$^a$ | (156.9) | 4.6$^{bc}$ | (20.9) | 9.85$^b$ | (108.0) |
| T$_3$ | 5.2$^b$ | (24.7) | 8.2$^b$ | (67.1) | 11.03$^b$ | (122.9) |
| T$_4$ | 12.3$^a$ | (150.3) | 6.8$^{bc}$ | (48.5) | 11.83$^{ab}$ | (140.5) |
| T$_5$ | 12.2$^a$ | (152.9) | 6.9$^{bc}$ | (50.0) | 11.23$^b$ | (127.1) |
| T$_6$ | 6.5$^b$ | (36.5) | 4.7$^{bc}$ | (22.1) | 9.13$^b$ | (88.4) |
| T$_7$ | 5.4$^b$ | (26.8) | 4.0$^c$ | (16.3) | 8.95$^b$ | (81.8) |

Note: Weed data subject to $\sqrt{X+0.5}$ transformation. Figures in parenthesis are original values.

Pooled means followed by the same letters are not significantly different according to Tukey's Honest Significant Difference (P = 0.05).

control treatments, treatment T7 recorded lowest weed density (8.95) at harvest stage which was at par with other weed control treatments except T4.

## Growth and yields

Significant influence of various herbicidal weed control treatments on plant growth and yield parameters were recorded as a result of their impact on weed population and their growth as compared to no weeding control (T1). Tallest plant height (50.98 cm) was recorded under T2 which was at par with rest of the herbicidal weed control treatments except control (Table 3). Maximum number of branches per plant (8.60) was recorded under T2 which was at par with rest of the herbicidal weed control treatments. The lowest number of branches per plant (6.71) was recorded under control (T1). The no weeding control treatment recorded significantly lower number of pods per plant (17.23) as compared to other weed control treatments. Significantly higher number of pods per plant (43.43) was recorded under treatment T2 which was on par with treatments T6 and T7 with 41.08 and 39.98 pods per plant, respectively. Similarly, all the weed control treatments produced higher number of seeds per pods as compared to control (2.17). Maximum number of seeds per pod (2.56) was recorded under treatment T2 which was at par with treatments T4, T6 and T7 and significantly higher as compared to rest of the weed control treatments. Maximum crop biomass and straw yield (4513 kg ha$^{-1}$ and 2951 kg ha$^{-1}$) was recorded under treatment T6 which was on par with rest of the herbicidal weed control treatments and hand weeding and significantly superior over treatment T1 and T3. Maximum seed yield (1567 kg ha$^{-1}$) and harvest index (37.37%) was recorded under treatment T2 which was at par with treatments T6, T7 and T5 and significantly higher as compared to treatments T1, T3 and T4.

## Weed control efficiency, weed biomass and weed index

Among different treatments, maximum WCE in terms of population at harvest stage (72.83%) has been recorded under T7 (Table 4) followed by T6 (70.66%). Better weed control under herbicide applied plots were attributed to weed control and improved crop performance due to sequential application of herbicides in combination of pendimethalin + glyphosate as pre-emergence followed by post-emergence application of propaquizafop+ chlorimuron ethyl (T7) or imazethapyr (T6) at 30 DAS. Among different treatments the lowest weed biomass (562 kg ha$^{-1}$) was recorded under T2 (hand weeding) which was statistically at par with treatment T7 (814 kg ha$^{-1}$). Maximum weed biomass 3850 kg ha$^{-1}$ was recorded (Table 3) under control (T1). Maximum WCI in terms of biomass (81.95%) has been recorded under T2 followed by treatment T6 (81.52%). Lower weed index (0.32%) was recorded under treatment T6.

**Table 3. Effect of different weed management practices on crop growth, yield attributes and yields.**

| Treatment | Plant height (cm) | Branches plant$^{-1}$ | Pods plant$^{-1}$ | Seeds pod$^{-1}$ | Biological yield (kgha$^{-1}$) | Straw yield (kg ha$^{-1}$) | Grain yield (kg ha$^{-1}$) | Harvest index (%) |
|---|---|---|---|---|---|---|---|---|
| T$_1$ | 38.90[b] | 6.71[b] | 17.23[c] | 2.17[b] | 1181[c] | 810[c] | 371[c] | 31.47[a] |
| T$_2$ | 50.98[a] | 8.60[a] | 43.43[a] | 2.56[a] | 4437[a] | 2870[a] | 1567[a] | 35.37[a] |
| T$_3$ | 45.39[ab] | 8.06[a] | 34.63[b] | 2.26[b] | 2946[b] | 2058[b] | 888[b] | 30.53[a] |
| T$_4$ | 50.42[a] | 8.46[a] | 37.90[ab] | 2.35[ab] | 3660[ab] | 2463[ab] | 1196[ab] | 32.80[a] |
| T$_5$ | 49.30[a] | 8.17[a] | 35.56[b] | 2.28[b] | 3853[ab] | 2503[ab] | 1350[a] | 35.07[a] |
| T$_6$ | 49.89[a] | 8.57[a] | 41.08[ab] | 2.39[ab] | 4513[a] | 2951[a] | 1562[a] | 35.13[a] |
| T$_7$ | 49.27[a] | 8.47[a] | 39.98[ab] | 2.36[ab] | 4236[a] | 2805[ab] | 1431[a] | 33.97[a] |

Pooled means followed by the same letters are not significantly different according to Tukey's Honest Significant Difference (P = 0.05).

**Table 4. Effect of different weed management practices on weed biomass, weed control efficiency (WCE) and weed index in soybean at harvest.**

| Treatment | Weed control efficiency Population (%) | Weed dry biomass (kg ha$^{-1}$) | Weed control efficiency Biomass (%) | Weed index (%) |
|---|---|---|---|---|
| T$_1$ | - | 3848$^a$ | - | 75.9 |
| T$_2$ | 66.6 | 562$^d$ | 82.0 | 0.0 |
| T$_3$ | 59.2 | 1880$^b$ | 55.4 | 42.8 |
| T$_4$ | 55.3 | 1758$^b$ | 55.8 | 23.0 |
| T$_5$ | 59.7 | 1614$^{bc}$ | 61.8 | 12.9 |
| T$_6$ | 70.7 | 814$^d$ | 81.5 | 0.3 |
| T$_7$ | 72.8 | 1425$^c$ | 67.3 | 8.0 |

Pooled means followed by the same letters are not significantly different according to Tukey's Honest Significant Difference (P = 0.05).

Maximum weed index (74.55%) was recorded under control (T1), indicating the severity of weed infestation and competition under Vertisols which ultimately resulted in significant yield reduction under no weeding control (T1) treatment.

## Nutrient uptake and nutrient use efficiency

Nutrient uptake in grain and straw was also found to be significantly influenced in response of different herbicidal treatments in soybean (Table 5). The highest total nutrient uptake (235.27 kg ha$^{-1}$) was recorded under T6 where sequential per-emergence and post-emergence herbicides was applied and followed by T2 (hand weedings at 20 and 40 DAS) (232.19 kg ha$^{-1}$) which was at par with T2 (hand weedings only) T4, T5 and T7 treatments while lowest nutrient (NPK) uptake in soybean was noticed in control plots. Furthermore, the results revealed that pre-emergence of herbicides (T3) of also performed better in respect to nutrient uptake as compared to control plots.

Agronomic use efficiency (AUE) for N, P and K also significantly increased under different herbicidal treatment whereas physiological use efficiency (PUE) (Table 6) was not influenced by the applied herbicidal weed management practices in soybean. The higher AUE for N (52.23 kg kg$^{-1}$of nutrient applied) was recorded in T2 treatment followed by T6 (52.09 kg kg$^{-1}$of nutrient applied), T7 (547.70 kg kg$^{-1}$of nutrient applied) and T5 (45.01) kg kg$^{-1}$of nutrient applied) and similar trend was also followed in case of P and K. While, the lower AUE was noticed in control plots in respect to N, P and K.

**Table 5. Effect of different weed management practices on nutrient (NPK) uptake in soybean.**

| Treatment | Nutrient uptake Grain (kg ha$^{-1}$) | | | Nutrient uptake straw (kg ha$^{-1}$) | | | Total Nutrient uptake (kg ha$^{-1}$) | | | |
|---|---|---|---|---|---|---|---|---|---|---|
| | N | P | K | N | P | K | N | P | K | NPK |
| T$_1$ | 17.16$^d$ | 1.72$^c$ | 5.79$^d$ | 20.43$^c$ | 2.87$^b$ | 9.93$^c$ | 37.60$^c$ | 4.60$^c$ | 15.72$^c$ | 57.94$^c$ |
| T$_2$ | 76.10$^a$ | 7.52$^a$ | 25.46$^a$ | 77.10$^a$ | 9.38$^a$ | 36.61$^{ab}$ | 153.20$^a$ | 16.90$^a$ | 62.07$^a$ | 232.19$^a$ |
| T$_3$ | 41.62$^c$ | 4.12$^b$ | 13.58$^c$ | 53.01$^b$ | 7.28$^a$ | 26.67$^b$ | 94.64$^b$ | 11.42$^b$ | 40.26$^b$ | 146.32$^b$ |
| T$_4$ | 57.39$^{bc}$ | 5.82$^{ab}$ | 18.36$^{bc}$ | 68.03$^{ab}$ | 8.88$^a$ | 33.08$^{ab}$ | 125.43$^{ab}$ | 14.70$^{ab}$ | 51.44$^{ab}$ | 191.59$^{ab}$ |
| T$_5$ | 65.37$^{ab}$ | 6.68$^a$ | 23.27$^{ab}$ | 67.80$^{ab}$ | 8.05$^a$ | 32.79$^{ab}$ | 133.17$^a$ | 14.74$^{ab}$ | 56.07$^a$ | 204.00$^a$ |
| T$_6$ | 74.36$^{ab}$ | 7.41$^a$ | 26.26$^a$ | 80.59$^a$ | 9.34$^a$ | 37.29$^a$ | 154.95$^a$ | 16.75$^a$ | 63.55$^a$ | 235.27$^a$ |
| T$_7$ | 67.66$^{ab}$ | 6.60$^a$ | 22.93$^{ab}$ | 77.04$^a$ | 8.85$^a$ | 36.03$^{ab}$ | 144.90$^a$ | 15.45$^a$ | 58.97$^a$ | 234.19$^a$ |

Pooled means followed by the same letters are not significantly different according to Tukey's Honest Significant Difference (P = 0.05).

**Table 6. Agronomic and physiological efficiencies influenced by weed management practices in soybean.**

| Treatment | Agronomic use efficiency (kg kg$^{-1}$ of nutrient applied) | | | Physiological use efficiency (kg kg$^{-1}$ of nutrient uptake) | | |
|---|---|---|---|---|---|---|
| | N | P | K | N | P | K |
| T$_1$ | 12.37$^c$ | 6.18$^c$ | 9.27$^c$ | 9. 87$^a$ | 80.63$^a$ | 23.62$^a$ |
| T$_2$ | 52.23$^a$ | 26.11$^a$ | 39.17$^a$ | 10.22$^a$ | 93.23$^a$ | 25.31$^a$ |
| T$_3$ | 29.60$^b$ | 14.80$^b$ | 22.20$^b$ | 9.46$^a$ | 78.86$^a$ | 22.41$^a$ |
| T$_4$ | 39.89$^{ab}$ | 19.94$^{ab}$ | 29.92$^{ab}$ | 9.56$^a$ | 81.68$^a$ | 23.33$^a$ |
| T$_5$ | 45.01$^a$ | 22.50$^a$ | 33.76$^a$ | 10.12$^a$ | 91.60$^a$ | 24.07$^a$ |
| T$_6$ | 52.09$^a$ | 26.04$^a$ | 39.07$^a$ | 10.18$^a$ | 94.55$^a$ | 24.89$^a$ |
| T$_7$ | 47.70$^a$ | 23.85$^a$ | 35.77$^a$ | 9.90$^a$ | 93.27$^a$ | 24.37$^a$ |

Pooled means followed by the same letters are not significantly different according to Tukey's Honest Significant Difference (P = 0.05).

## Crop residue retention, carbon and nutrient recycling

As a result of better crop performance in terms of crop growth, more crop biomass, carbon and nutrient accumulation has been observed. Residue retention, carbon input and nutrient recycled in soybean was also significantly influenced under different herbicidal weed management treatments under CA, highest residue (885 kg ha$^{-1}$year$^{-1}$), carbon input (354 kg ha$^{-1}$year$^{-1}$), and nutrient recycled (24.17, 2.80 and 11.18 kg N, P and K ha$^{-1}$year$^{-1}$, respectively) was recorded under treatment T6 which was at par with treatment T2, T7, T5 and T4 and significantly higher as compared to treatmentsT1 and T3 during the course of study (Table 7).

## Correlation and regression relationship to yield and weed parameters

Pearson's correlation coefficients matrix (Table 8) revealed that grain yield was significantly (P<0.0001) related to WD60 (r = -0.84), WDH (r = -0.63), WDB (r = -0.52), WN (r = -0.56), WP (r = -0.59), WK (r = -0.66) and WI (r = -0.96). The bivariate linear regression study of grain yield on WCI biomass, N, P and K uptake showed the clear significant (P<0.0001) dependence with R$^2$ value of 0.53 (Fig 1), 0.99 (Fig 2), 0.95 (Fig 3) and 0.98 (Fig 4), respectively.

The stepwise multiple regression (model) analysis (Eq 1), indicated that some variables has truly influenced (significantly) the grain yield as compared to others. The fitted model showed that among the different parameters, only N and P uptake in grain, weed density at 20 DAS

**Table 7. Carbon and nutrient addition through residue recycling in soybean.**

| Treatment | Residue recycled (kg ha$^{-1}$year$^{-1}$) | Carbon input (kg ha$^{-1}$ year$^{-1}$) | Nutrient recycled (kg ha$^{-1}$ year$^{-1}$) | | |
|---|---|---|---|---|---|
| | | | N | P | K |
| T$_1$ | 243$^c$ | 97$^c$ | 6.13$^c$ | 0.86$^b$ | 2.97$^c$ |
| T$_2$ | 861$^a$ | 344$^a$ | 23.13$^a$ | 2.81$^a$ | 10.98$^{ab}$ |
| T$_3$ | 617$^b$ | 246$^b$ | 15.90$^b$ | 2.18$^a$ | 8.00$^b$ |
| T$_4$ | 739$^{ab}$ | 295$^{ab}$ | 20.41$^{ab}$ | 2.66$^a$ | 9.92$^{ab}$ |
| T$_5$ | 751$^{ab}$ | 300$^{ab}$ | 20.34$^{ab}$ | 2.41$^a$ | 9.84$^{ab}$ |
| T$_6$ | 885$^a$ | 354$^a$ | 24.17$^a$ | 2.80$^a$ | 11.18$^a$ |
| T$_7$ | 841$^{ab}$ | 336$^{ab}$ | 23.11$^a$ | 2.65$^a$ | 10.81$^{ab}$ |

Pooled means followed by the same letters are not significantly different according to Tukey's Honest Significant Difference (P = 0.05).

**Table 8. Pearson's correlation coefficients (N = 63) between yield and weed parameters.**

| Parameters | GY | WD20 | WD60 | WDH | WDB | WN | WP | WK | WI |
|---|---|---|---|---|---|---|---|---|---|
| **GY** | 1.00000 | -0.12846 (0.3157) | -0.84432 (< .0001) | -0.63054 (< .0001) | -0.51733 (< .0001) | -0.55532 (< .0001) | -0.59068 (< .0001) | -0.65661 (< .0001) | -0.96115 (< .0001) |
| **WD20** | | 1.00000 | 0.36129 (0.0036) | 0.48020 (< .0001) | 0.26826 (0.0335) | 0.28162 (0.0254) | 0.34194 (0.0061) | 0.28395 (0.0241) | 0.13302 (0.2987) |
| **WD60** | | | 1.00000 | 0.78447 (< .0001) | 0.63483 (< .0001) | 0.66898 (< .0001) | 0.71506 (< .0001) | 0.76218 (< .0001) | 0.81590 (< .0001) |
| **WDH** | | | | 1.00000 | 0.62353 (< .0001) | 0.64794 (< .0001) | 0.69424 (< .0001) | 0.71356 (< .0001) | 0.65138 (< .0001) |
| **WDB** | | | | | 1.00000 | 0.99688 (< .0001) | 0.98677 (< .0001) | 0.96859 (< .0001) | 0.52629 (< .0001) |
| **WN** | | | | | | 1.00000 | 0.98894 (< .0001) | 0.97746 (< .0001) | 0.56251 (< .0001) |
| **WP** | | | | | | | 1.00000 | 0.98015 (< .0001) | 0.59566 (< .0001) |
| **WK** | | | | | | | | 1.00000 | 0.65902 (< .0001) |
| **WI** | | | | | | | | | 1.00000 |

GY, Grain yield; WD20, Weed density at 20 DAS; WD60, Weed density at 60 DAS; WDH, Weed density at harvest; WDB, Weed dry biomass; WN, N uptake in weed; WP, P uptake in weed; WK, K uptake in weed; WI, Weed index.

Values in the parenthesis is the actual probability level of significance; if the value is ≤0.05 or 0.01, then r (the correlation coefficient) ≠ 0 at P = 0.05 or 0.01.

and K uptake in weed were actual predictor for grain yield.

$$GY = 0.19 **xNG + 0.21 **xPG - 0.04 **xWD20 + 0.01 * xWK, (R^2 = 0.99; P < 0.0001) \quad \text{(Eq 1)}$$

($^*$and$^{**}$ indicate the parameters estimate are significant at P = 0.05 and P = 0.01)

## Discussion

### Weed flora and weed density

In the experimental field monocot weeds were dominant and recorded 62.3% density as compared of dicot weeds which contribute to the extent of 35% and sedges 2.6%. Among monocot weed species *Echinochloa colonum* (L.) was the most dominant weed followed by *Dichanthium annulatum* forssk. Among dicot weeds, *Acalypha indica* L. was the dominant one. Similarly, weed flora in soybean cultivated in Vertisols were also reported by Malik *et al*. [47]. Significant effect of different weed control treatments on density of dominant weeds at different growth stages was recorded under CA (Table 2). Application of glyphosate alone and in combination with pendimethalin as pre-emergence application has effectively controlled emerged weeds as well as do not allow fresh flush during initial crop growth phase and also reported significant reduction in weed density of all the species due to pre-emergence application of herbicides as compared to weedy check [47]. Significantly lower weed densities in all the treatments as compared to no weeding control may be attributed to the action of different combinations of pre- and post-emergence herbicide treatments applied to soybean crop, whereas in case of pendimethalin alone treatment the number of weeds increased substantially towards maturity of crop [48]. Application of pendimethalin resulted in inhibition of microtubule formation in cells of sensitive weeds, thus interfere with the cell division process of germinating seeds. This resulted in restricted cell division and reduced growth of the emerging weed seedling eventuated to lack of food reserves and ultimately results in death of germinating seedlings.

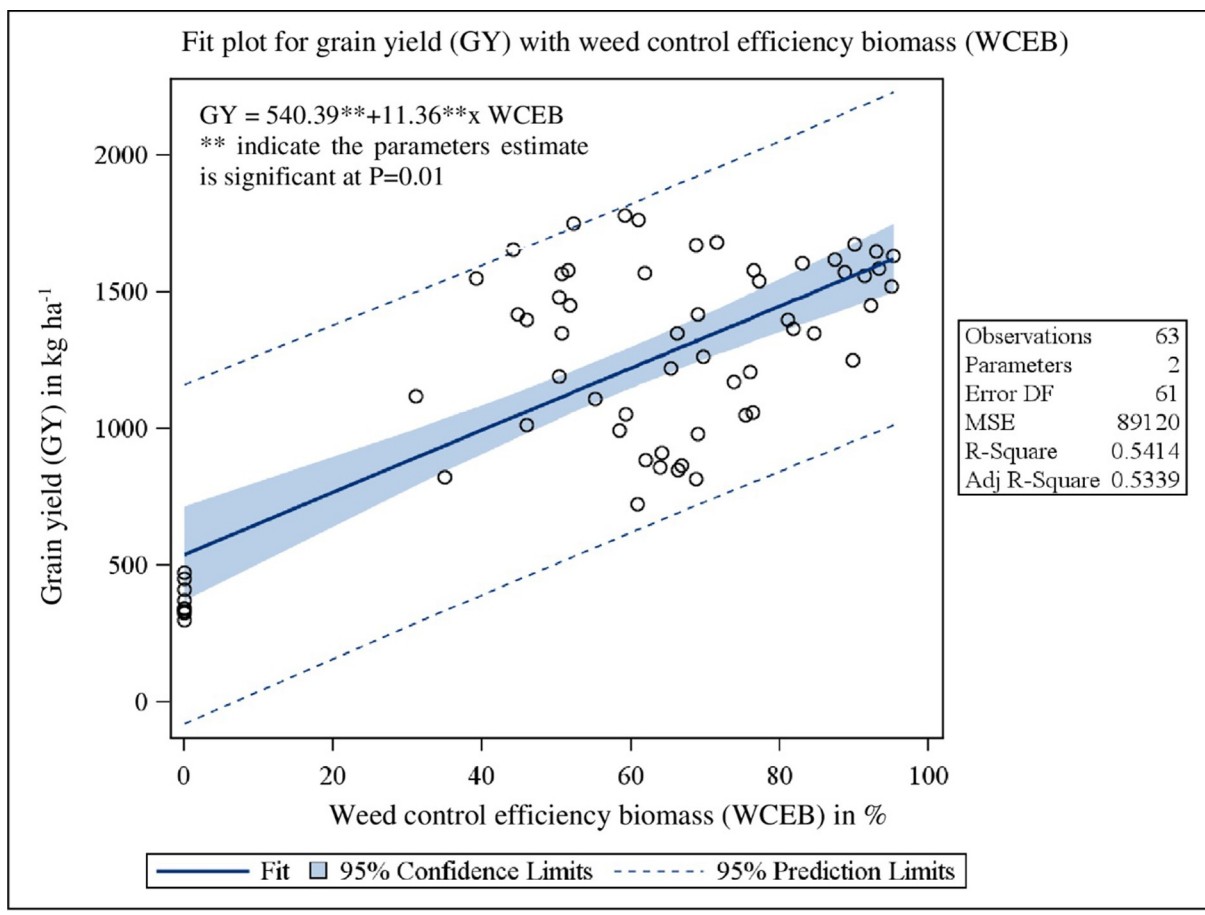

**Fig 1. Relationship between grain yield (GY) and weed control efficiency biomass (WCEB) under herbicidal weed management practices in soybean.**

Inhibition of major enzymatic processes responsible for protein synthesis *viz.*, acetolactate synthase (ALS) or acetohydroxy acid synthase (AHAS) in broad leaf weeds due to post emergence application of imazethapyr at 3–4 leaf stage is responsible for control of these weeds due to starvation [48]. Inhibition of ALS enzyme mediated processes resulted in gradual killing of target weeds in mixed population. Similar findings were also reported by Kumar *et al.* [49]. The lowest biomass, weed index and highest WCI was found in treatment T7 involving pre-emergence application of pendimethalin at 1 kg ai ha$^{-1}$ + glyphosate at 1 kg ai ha$^{-1}$ followed by imazethapyr 0.100 kg ai ha$^{-1}$ as PoE. This may be attributed mainly to better weed control as a result of post-emergence application of imazethapyr applied at 3–4 leaf stage [50].

## Weed control efficiency, weed biomass and weed index

Season long weed control and higher WCE in terms of population due to sequential application of pre- and post-emergence herbicides gave better results as compared to their individual applications. Thus, resulted in lower weed biomass, improved crop performance and lower weed index under these treatments [22] also reported greater efficacy at lower cost with post-emergence application of herbicides and these findings were also in close conformity of the same. Higher WCI underhand weeding treatment in terms of biomass as compared to WCI in terms of population may be attributed to the fact that a higher number of weeds emerged

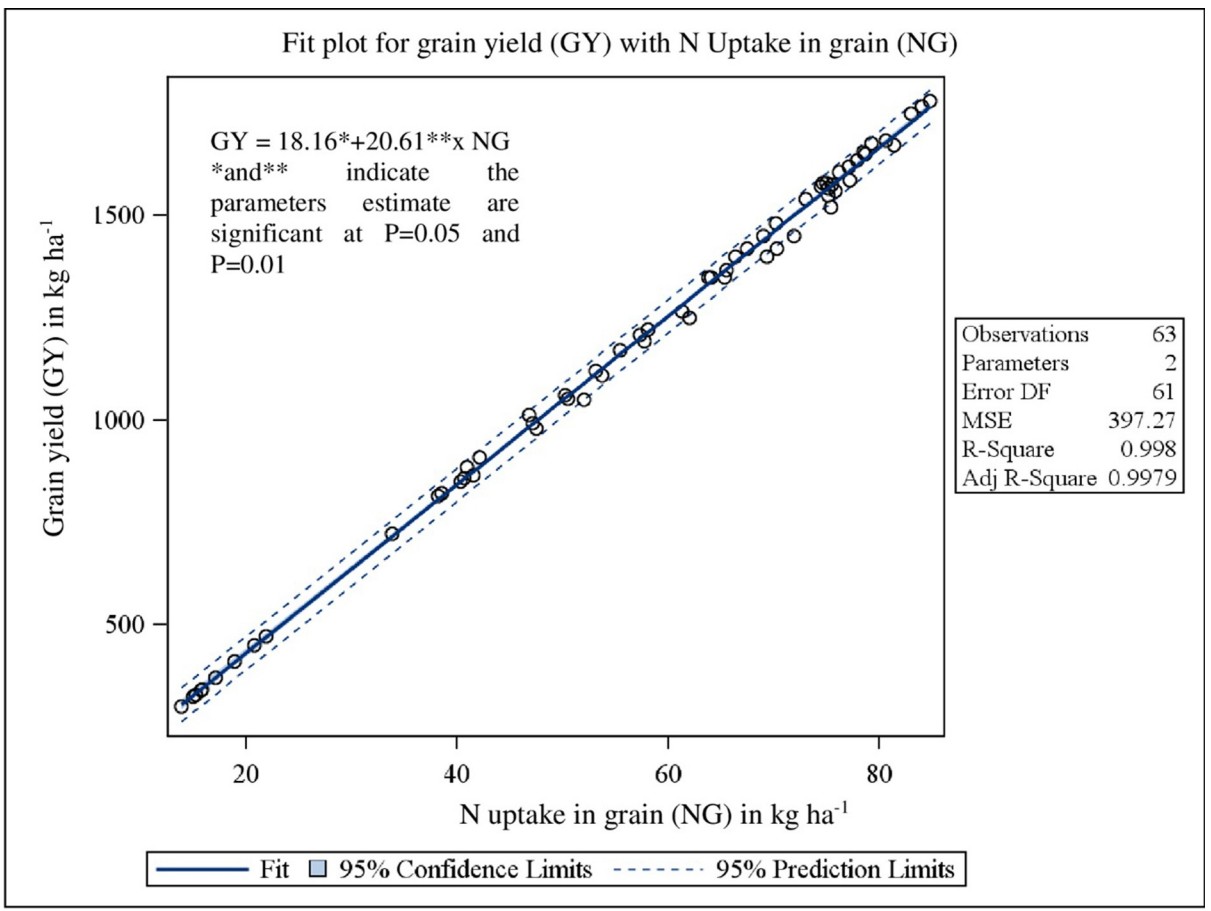

**Fig 2. Relationship between grain yield (GY) and N uptake in grain (NG) under herbicidal weed management practices in soybean.**

towards maturity phase of the crop which could not attain higher biomass as compared to weeds which emerge earlier during the crop cycle in other treatments. These results were in close conformity with that of reported by Prachand [50]. As a result of increased crop yield under different weed control treatments, lower weed index was observed in treated plots which indicate the severity of weed infestation and competition by weeds in soybean under Vertisols which resulted in significant yield reduction under no weeding control treatment.

## Growth and yields

Significant response of various herbicidal treatments on plant growth and yield parameters of soybean may be attributed to availability of favourable environment for crop as a result of proper weed control, which reduces the competition of crop with weeds for space, air, sunlight, moisture and nutrients which is reflected in terms of significantly higher seed yield, (1567 kg ha⁻¹) and harvest index (37.37%). Prachand *et al.* [50] also reported increased yield and harvest index in soybean as a result of weed management due to better crop growth and yield parameters. Muoni *et al.* [51] suggested that yield losses due to crop-weed competition are reduced as a result of decrease in weed density. Besides reducing the weed density and growth the herbicidal weed control treatments also resulted in reduced the nutrient depletion by weeds and thereby increasing the nutrient uptake by crop by providing congenial environment for crop growth and development which helped the soybean crop to produce a greater number of pods

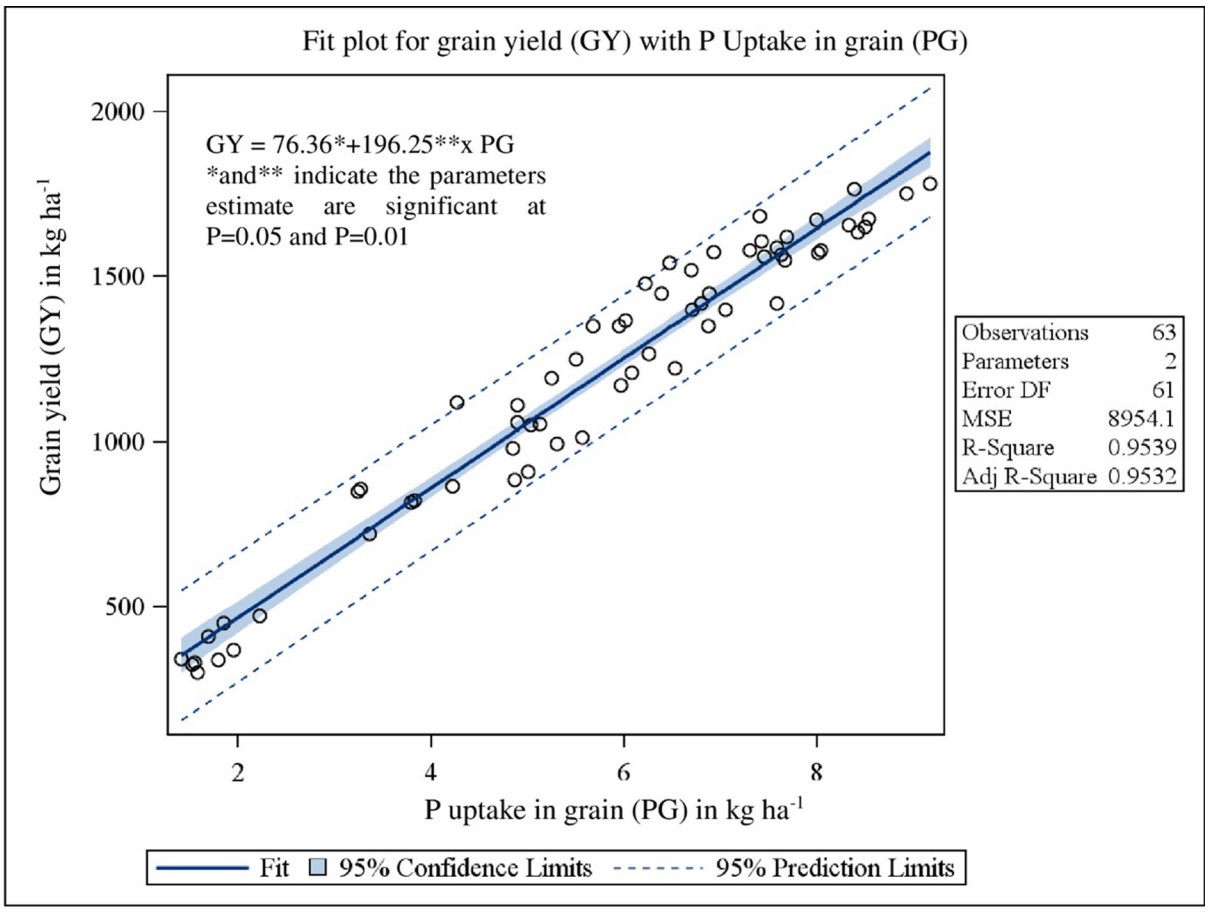

**Fig 3. Relationship between grain yield (GY) and P uptake in grain (PG) under herbicidal weed management practices in soybean.**

plant$^{-1}$ and seeds plant$^{-1}$. The improved growth and yield parameters ultimately reflected in terms of higher crop yield in herbicidal weed control treatments as compared to no weeding control and were at par with hand weeding treatment [52] also reported improvement in yield and economical parameters as a result of better weed control with herbicidal weed management practices in soybean.

## Nutrient uptake and nutrient use efficiency

Weeds are more efficient competitors for nutrients, water, space utilization as compared to soybean, which ultimately have negative effects on crop yield and nutrient uptake by crop [52]. Significant response of different herbicidal treatments on nutrient uptake in seed and straw as compared to nutrient uptake in seed and straw under control was recorded in treatments (Table 5). This might be due to better crop growth which resulted in higher grain and straw yield in the respective treatments and ultimately resulted in higher nutrient uptake due to availability of proper space and light to crop as a result of increased weed control in herbicidal treatments during critical periods of crop growth [53]. This may also be attributed to the impact of lower competition under herbicide applied plots, which as a result of better weed suppression resulted in improved canopy development and increased the nutrient uptake. Increase in nutrient uptake (NPK) also suggests that NUE was high under the given treatment due to reduced crop-weed competition and also increase the crop yield due to less weed

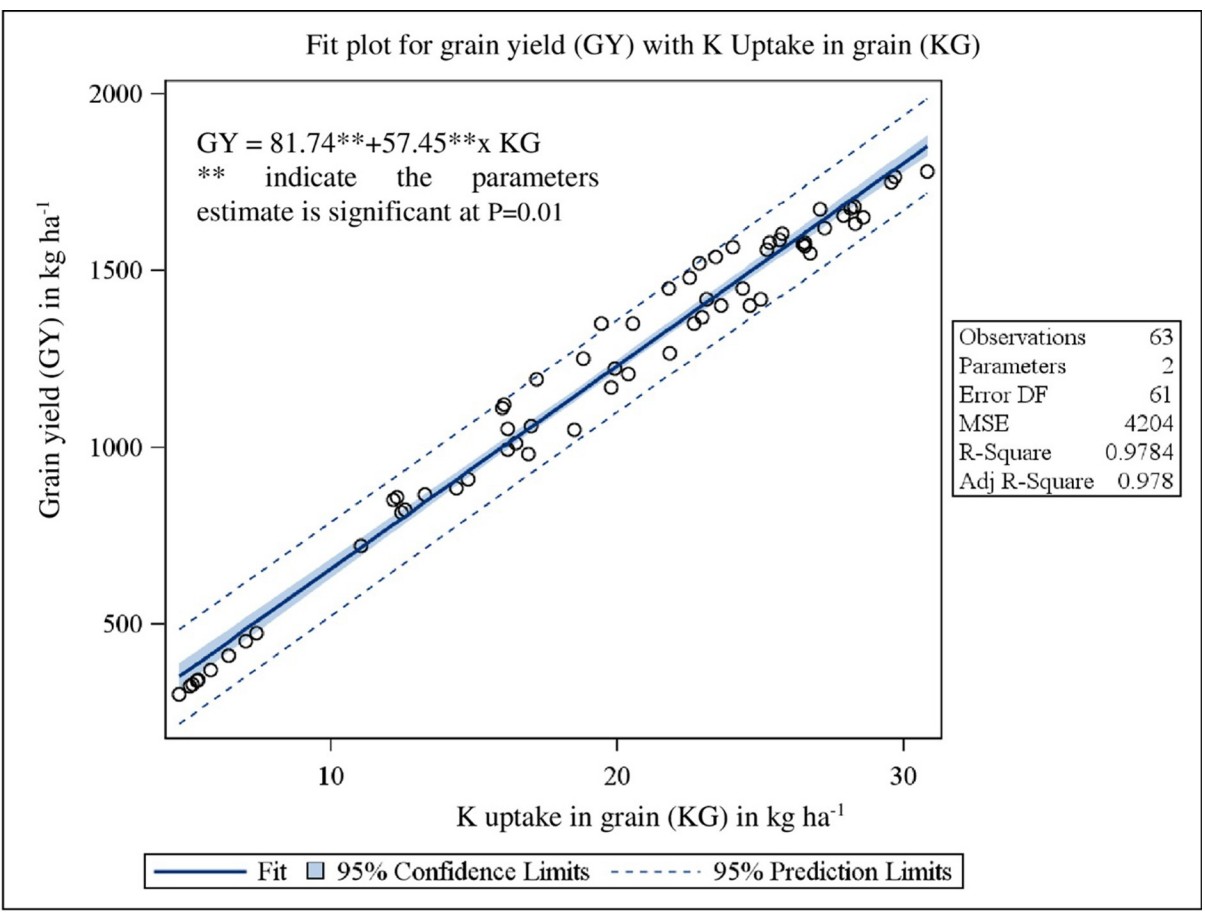

**Fig 4. Relationship between grain yield (GY) with K uptake in grain (KG) under herbicidal weed management practices in soybean.**

population [51,53]. Significantly higher AUE (Table 6) were recorded under various weed control treatments, this may be attributed to the impact of weed control treatments in terms of better crop growth and reduced crop weed competition, which resulted in greater nutrient uptake under these treatments as compared to control. On the contrary, PUE was not influenced by the applied herbicidal weed management in soybean crops during the experimentation. Chauhan *et al.* [22] also reported similar observations and suggested that herbicidal weed management practices under CA are very much suitable for obtaining higher NUE and higher yield. However, PUE of N, P, and K were non-significant, indicating that the biomass production is influenced by the amount of nutrients absorbed by the soybean crop, but the composition of the biomass is not influenced significantly by the amount of nutrients absorbed by the crop [54].

## Crop residue retention, carbon and nutrient recycling

Crop residue retention is core principle of CA production system through its influence on the soil-plant inter-relationship, which not only helps in nutrient recycling in long term [55,56], but also helps in soil health improvement [57] and reduction in soil degradation [58]. Significant influence of herbicidal weed management treatments in terms of crop biomass production, carbon and nutrient recycling under CA is due to retention of higher residue biomass under different treatments due to differential crop growth, which resulted in accumulation of

more crop biomass, carbon and nutrient under different treatments [59]. Therefore, retention of higher amount of residue in treatments producing more residues as a result of better growth resulted in accumulation of higher amount of carbon in soybean based agricultural production systems and increased nutrient recycling [60,61]. As a result of improved crop performance; the crop acquires a major portion of resources that are required for growth and development of plant which ultimately resulted in increased residue, carbon and nutrient recycling [62].

### Correlation and regression relationship to yield and weed parameters

The correlation matrix represents the interrelationships among different variables, to which the crop yield is negatively correlated with all other yield affecting parameters *viz*., density of the weeds, weed biomass, weed nutrient uptakes and weed index (Table 8). Pearson's correlation coefficients matrix revealed that grain yield was significantly ($P<0.0001$) related to all the parameters except weed density at 20 DAS. The bivariate linear regression study of grain yield on weed control efficiency (biomass), nutrient uptake (N, P and K) in grain showed the clear significant ($P<0.0001$) dependence with good fit $R^2$ values. Stepwise multiple regression analysis (Eq 1) showed that N and P uptake in grain, weed density at 20 DAS and K uptake in weed was true significant predictor for grain yield.

## Conclusion

Adoption of zero-till farming leads to increased weed infestation under CA due to the predominance and accumulation of weed seeds on the upper surface of the soil. Under CA, reliance on herbicidal weed management is the only feasible way to deal with this problem. However, a single herbicide may not be enough to provide season-long control of a diverse range of weeds Based on our salient findings, it can be concluded that under a CA-based soybean production system, weeds can be effectively managed through the utilization of different herbicides in suitable combinations applied at the desired rate at the right time. Sequential application of glyphosate at 1 kg ai ha[-1] along with pendimethalin at 1 kg ai ha[-1] as PE, followed by PoE application of imazethapyr at 100 g ai ha[-1] at 30 DAS, proved to be the most cost-effective method of weed control and crop productivity under CA. The higher residue retention, carbon input, and nutrient recycling in soybean were also noticed with herbicidal weed management under CA. However, future field studies should be conducted on different combinations of suitable herbicides to provide a more accurate recommendation for managing weeds under CA.

## Acknowledgments

Authors acknowledge ICAR-Indian Institute of Soil Science, Bhopal, India for providing technical assistance during the course of study.

## Author Contributions

**Conceptualization:** A. K. Vishwakarma, Bharat Prakash Meena, A. K. Biswas, R. S. Chaudhary, Ashok K. Patra.

**Data curation:** A. K. Vishwakarma, Bharat Prakash Meena, Hiranmoy Das, K. Bharati, R. S. Chaudhary.

**Formal analysis:** A. K. Vishwakarma, Bharat Prakash Meena, Pramod Jha, R. S. Chaudhary.

**Funding acquisition:** Pramod Jha, A. K. Biswas, K. M. Hati.

**Investigation:** A. K. Vishwakarma, Bharat Prakash Meena, K. Bharati, K. M. Hati.

**Methodology:** A. K. Vishwakarma, K. M. Hati.

**Project administration:** A. K. Biswas, R. S. Chaudhary, Ashok K. Patra.

**Resources:** Bharat Prakash Meena, Pramod Jha, K. Bharati, R. S. Chaudhary, A. O. Shirale, Priya P. Gurav, Ashok K. Patra.

**Software:** Hiranmoy Das.

**Supervision:** A. O. Shirale, B. L. Lakaria, Priya P. Gurav, Ashok K. Patra.

**Validation:** A. K. Biswas, K. Bharati, K. M. Hati.

**Visualization:** A. K. Vishwakarma, Pramod Jha, A. K. Biswas, K. Bharati, B. L. Lakaria, Priya P. Gurav.

**Writing – original draft:** A. K. Vishwakarma, Bharat Prakash Meena, R. S. Chaudhary, A. O. Shirale, B. L. Lakaria.

**Writing – review & editing:** A. K. Vishwakarma, Bharat Prakash Meena, Hiranmoy Das, A. K. Biswas, R. S. Chaudhary, A. O. Shirale, B. L. Lakaria, Priya P. Gurav, Ashok K. Patra.

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
