## [Decision Letter · Decision Letter 0]

29 Jul 2022

PONE-D-22-19003Impact of sequential herbicides application on crop productivity, weed and nutrient dynamics in soybean under conservation agriculture in Vertisols of Central IndiaPLOS ONE

Dear Dr. Meena,

Thank you for submitting your manuscript to PLOS ONE. After careful consideration, we feel that it has merit but does not fully meet PLOS ONE’s publication criteria as it currently stands. Therefore, we invite you to submit a revised version of the manuscript that addresses the points raised during the review process. Please submit your revised manuscript by Sep 12 2022 11:59PM. If you will need more time than this to complete your revisions, please reply to this message or contact the journal office at plosone@plos.org. Please include the following items when submitting your revised manuscript:A rebuttal letter that responds to each point raised by the academic editor and reviewer(s). You should upload this letter as a separate file labeled 'Response to Reviewers'.A marked-up copy of your manuscript that highlights changes made to the original version. You should upload this as a separate file labeled 'Revised Manuscript with Track Changes'.An unmarked version of your revised paper without tracked changes. You should upload this as a separate file labeled 'Manuscript'.

We look forward to receiving your revised manuscript.

Kind regards,

Mohan Lal Dotaniya, Ph.D.

Academic Editor

PLOS ONE

Journal Requirements:

"Authors acknowledge ICAR-Indian Institute of Soil Science, Bhopal India for providing financial and technical assistance during the course of study."

Additional Editor Comments:

Carefully incorporate the reviewers comments and suggestions. A letter of correction/modification should be attached during submission.

Reviewers' comments:

Reviewer's Responses to Questions

**Comments to the Author**

1. Is the manuscript technically sound, and do the data support the conclusions?

Reviewer #1: Yes

Reviewer #2: Partly

Reviewer #3: Yes

Reviewer #4: Yes

2. Has the statistical analysis been performed appropriately and rigorously? 

Reviewer #1: Yes

Reviewer #2: Yes

Reviewer #3: Yes

Reviewer #4: Yes

3. Have the authors made all data underlying the findings in their manuscript fully available?

Reviewer #1: Yes

Reviewer #2: Yes

Reviewer #3: Yes

Reviewer #4: Yes

4. Is the manuscript presented in an intelligible fashion and written in standard English?

Reviewer #1: Yes

Reviewer #2: Yes

Reviewer #3: Yes

Reviewer #4: Yes

5. Review Comments to the Author

Reviewer #1: Comments for Authors

• Abstract written very well however some minor correction need to be incorporated like how much per-chant age increase in crop productivity under the best system.

• Introduction: Introduction is very well written indicating about background and hypothesis of study about weed management under conservation agriculture. However, it can be condensed as it is very largely elaborated.

• Materials and methods: This section described methods of all parameters and can be included some spelling mistakes with some minor correction and modifications.

• Whether the crop variety (JS-335, soybean) used same in all years? and author performed the field study under rainfed ecosystem particularly under conservation agriculture, whether the used crop variety was screen out for the conservation agriculture?

• Soybean crop is very susceptible to insect and pest attack, what type plant protection measures were used to check the pest problem.

• Nitrogen, phosphorus and potassium can be written as N, P and K as it is well understood, and no need to write again and again.

• Recommended rate of fertilizer (30 kg N+ 60 kg P2O5 +40 kg K2O ha-1) has been drilled uniformly at the time of sowing with ‘happy seeder’ as basal application, whether, it was drilled or broadcast or sprayed, check it again ?.

• Results: Tables and figures were explained very well in result section, some minor modification is needed to be corrected.

• “Please check the Latin names of the weeds?.

• Discussion: I appreciate the discussion part, the way author discussed the results with other findings is quite good. Only I have a small query that author about the 30% residue level in all the plots irrespective of herbicidal treatments, whether it was ‘anchored’ residue or residue ‘incorporation’, please mentioned it clearly.

• Conclusion is written well although there is scope to improve it in briefly as per salient achievement of the study.

• We concluded….. replace with it can be concluded.

• References: All the cited references in text are in reference list and also as per the journal’s format, but please check it carefully again.

• English grammar and punctuation is very good throughout the manuscript and the study accompanying results have merit but minor revisions are needed to make this manuscript worthy of publication.

Reviewer #2: Authors have attempted a good piece of work for continuous three years to manage the weeds in soybean under CA.

However, the manuscript nee a major revision,

The conclusion part needs to be rewritten given the emphasis on the findings.

In the figures, y-axis (yield parameters) values need to be checked and changed accordingly.

The rest of the comments have been appended to the body of the text.

Thanks

Regards

Reviewer #3: Dear Editor,

Researchers have conducted study on to evaluate the expediency of different herbicides and their sequential applications under CA. The manuscript is well written, explained and discussed well, however, I am having few queries and should be incorporated it in manuscript before final decision.

1. The study was conducted for 3 years and data related to weed density, weed dry matter and grain yield was provided as a mean or average of three years, isn't it? You have rightly mentioned in the introduction section that the effect of CA is visible in longer run, here, I want to see the year wise data of weed density, weed dry matter, species wise difference and grain yield for year wise. I believe it will improve the quality of manuscript and easy to understandable for international readers. Kindly incorporate it?

2. Methodology of agronomic efficiency and PUE is missing in respective section, incorporate it?

3. Control plot- you mean herbicide was not applied in control treatment or nutrients were not applied, elaborate it?

Reviewer #4: There are few corrections/suggestions as given below:

• Scientific names of few weed spp. need to be rechecked.

• There are few typological errors

• Few new and relevant references can be added, if possible

• There are few errors occurred during conversion to PDF.

6. PLOS authors have the option to publish the peer review history of their article (what does this mean?). If published, this will include your full peer review and any attached files.

Reviewer #1: **Yes: **Dr ASHA RAM

Reviewer #2: No

Reviewer #3: **Yes: **Dr. B. Lal, Senior Scientist, ICAR-Indian Institute of Pulses Research, Regional Research centre, Bikaner

Reviewer #4: No

---

## [Author Response · Author response to Decision Letter 0]

12 Sep 2022

Reviewer 1

1. Comment: Abstract written very well however some minor correction need to be incorporated like how much per-chant age increase in crop productivity under the best system.

Response : Agreed and complied,

Thanks for appreciating of our efforts. 

Incorporated in the revised manuscript as per suggestions. 

2. Comment: Introduction: Introduction is very well written indicating about background and hypothesis of study about weed management under conservation agriculture. However, it can be condensed as it is very largely elaborated. 

Response : Thank you very much for appreciating of our efforts.

3. Comment: Materials and methods: This section described methods of all parameters and can be included some spelling mistakes with some minor correction and modifications.

Response : Agreed and corrected all the mistakes and incorporated in the revised manuscript as per suggestions. 

4 Comment: Whether the crop variety (JS-335, soybean) used same in all years? and author performed the field study under rainfed ecosystem particularly under conservation agriculture, whether the used crop variety was screen out for the conservation agriculture?

Response : Yes, same varieties were used for conducting all season experiment and variety and included in the material and method section of the manuscript.

5 Comment: Soybean crop is very susceptible to insect and pest attack, what type plant protection measures were used to check the pest problem.

Response : Imidacloprid and thiamethoxam were sprayed to control aphids, thrips and whiteflies while Emamectin benzoate was used to control lepidopteran pests.

6 Comment: Nitrogen, phosphorus and potassium can be written as N, P and K as it is well understood, and no need to write again and again.

Response : Thank you very much for your valuable suggestions on our manuscript 

Agreed and complied.

7 Comment: Recommended rate of fertilizer (30 kg N+ 60 kg P2O5 +40 kg K2O ha-1) has been drilled uniformly at the time of sowing with ‘happy seeder’ as basal application, whether, it was drilled or broadcast or sprayed, check it again ?.

Response : Agreed, complied and corrected as per suggestions.

8 Comment: Results: Tables and figures were explained very well in result section, some minor modification is needed to be corrected.

Response : Agreed and complied,

Thanks for appreciating of our efforts Incorporated as per suggestions. 

9 Comment: Please check the Latin names of the weeds?.

Response : Thank you much for pointing out. All the Latin names of the weeds are checked and corrected as per suggestions.

10 Comment: Discussion: I appreciate the discussion part, the way author discussed the results with other findings is quite good. Only I have a small query that author about the 30% residue level in all the plots irrespective of herbicidal treatments, whether it was ‘anchored’ residue or residue ‘incorporation’, please mentioned it clearly. 

Response : Thank you very much for appreciating our efforts.

11 Comment: Conclusion is written well although there is scope to improve it in briefly as per salient achievement of the study.

Response : Thank you very much for your positive comments on our manuscript. 

Agreed, the conclusion revised as per suggestions. 

12. Comment: We concluded….. replace with it can be concluded.

Response : Agreed, replaced as per suggestions.

13 Comment: References: All the cited references in text are in reference list and also as per the journal’s format, but please check it carefully again.

Response : Checked all the references and cited as per Journal format.

14 Comment: English grammar and punctuation is very good throughout the manuscript and the study accompanying results have merit but minor revisions are needed to make this manuscript worthy of publication.

Response : Agreed, the grammar and punctuation again also checked and corrected wherever required. Thanks for appreciating.

Reviewer #2:

1 Comment: Authors have attempted a good piece of work for continuous three years to manage the weeds in soybean under CA.

However, the manuscript need a major revision,

Response : Agreed and complied,

Thanks for appreciating of our efforts. The per-chant increase in crop productivity under the best is incorporated as per suggestions. 

2 Comment: The conclusion part needs to be rewritten given the emphasis on the findings.

Response : Thank you very much for your positive comments on our manuscript. Based on your comments, we have discussed and revised the contents of our conclusion. The purpose is to make our conclusions emphasize the scientific value of our research and highlight the innovation and novelty of our research.

3 Comment: In the figures, y-axis (yield parameters) values need to be checked and changed accordingly.

Response : Agreed and checked as per suggestions.

4 Comment: The rest of the comments have been appended to the body of the text.

Response : All the comments given in the manuscript are addressed and corrected in revised manuscript 

Reviewer #3:

1 Comment: Researchers have conducted study on to

evaluate the expediency of different herbicides and their sequential applications under CA. The manuscript is well written, explained and discussed well, however, I am having few queries and should be incorporated it in manuscript before final decision.

Response : Thank you very much for appreciating of our efforts. And all the queries are checked and corrected as per given suggestions. 

2 Comment: The study was conducted for 3 years and data related to weed density, weed dry matter and grain yield was provided as a mean or average of three years, isn't it? You have rightly mentioned in the introduction section that the effect of CA is visible in longer run, here; I want to see the year wise data of weed density, weed dry matter, species wise difference and grain yield for year wise. I believe it will improve the quality of manuscript and easy to understandable for international readers. 

Kindly incorporate it?

Response : Thank you very much for your valuable comments on our manuscript. We carefully and detailed studied of the given comments. However, at this junction it may not be possible to incorporate yearly data as it will change the basic framework of the manuscript.

3 Comment: Methodology of agronomic efficiency and PUE is missing in respective section, incorporate it?

 Response : Thank you very much for your valuable comments on our manuscript and Incorporated the missing methodology of agronomic efficiency and PUE in the revised manuscript. 

 Control plot- you mean herbicide was not applied in control treatment or nutrients were not applied, elaborate it?

Response : Agreed, herbicides not applied in control plots however, RDF was applied uniformly. 

Reviewer #4: 

Comment: Scientific names of few weed spp. need to be rechecked.

Response : All the scientific names are checked and corrected in the manuscript. 

 Comment: There are few typological errors•

Response : All the grammatical and typological errors are corrected in the revised manuscript. 

 Comment: Few new and relevant references can be added, if possible

Response : Agreed, included in the revised manuscript. 

Comment: There are few errors occurred during conversion to PDF.

Response : Thank you very much for your valuable suggestions on our manuscript. All the errors are checked and corrected as per given suggestions.

---

## [Decision Letter · Decision Letter 1]

12 Oct 2022

PONE-D-22-19003R1Impact of sequential herbicides application on crop productivity, weed and nutrient dynamics in soybean under conservation agriculture in Vertisols of Central IndiaPLOS ONE

Dear Dr. Meena,

Thank you for submitting your manuscript to PLOS ONE. After careful consideration, we feel that it has merit but does not fully meet PLOS ONE’s publication criteria as it currently stands. Therefore, we invite you to submit a revised version of the manuscript that addresses the points raised during the review process.

We look forward to receiving your revised manuscript.

Kind regards,

Mohan Lal Dotaniya, Ph.D.

Academic Editor

PLOS ONE

Journal Requirements:

Reviewers' comments:

Reviewer's Responses to Questions

**Comments to the Author**

1. If the authors have adequately addressed your comments raised in a previous round of review and you feel that this manuscript is now acceptable for publication, you may indicate that here to bypass the “Comments to the Author” section, enter your conflict of interest statement in the “Confidential to Editor” section, and submit your "Accept" recommendation.

Reviewer #1: All comments have been addressed

Reviewer #2: (No Response)

Reviewer #3: All comments have been addressed

2. Is the manuscript technically sound, and do the data support the conclusions?

Reviewer #1: Yes

Reviewer #2: Yes

Reviewer #3: Yes

3. Has the statistical analysis been performed appropriately and rigorously? 

Reviewer #1: Yes

Reviewer #2: Yes

Reviewer #3: Yes

4. Have the authors made all data underlying the findings in their manuscript fully available?

Reviewer #1: Yes

Reviewer #2: Yes

Reviewer #3: Yes

5. Is the manuscript presented in an intelligible fashion and written in standard English?

Reviewer #1: Yes

Reviewer #2: Yes

Reviewer #3: Yes

6. Review Comments to the Author

Reviewer #1: (No Response)

Reviewer #2: Manuscript has been significantly improved by the authors and it looks in good shape to accept for publication. The issue addressed in manuscript is very much pertaining to soybean cultivation in vertisol of Central India and similar agro-ecosystem. In recent past, many of the weed species has shown tolerant to ruling herbicides. Thus, information generated with this study will be useful to soybean growers and similar group crops (greengram and blackgram). However, in my previous review I have suggested to modify for figures Y axix, but I did not see it in present attached documents. This might be due to non-submission or not attached. Abbreviations and terminologies should be used appropriately throughout the manuscript. Spelling of scientific name of the weeds need to be re-checked. The weed density values need to be rounded off to nearest values, no need to present two digits after decimal. Weed index is yield dependent parameters so these may be presented beside yield. The suggested points may be looked critically. Rest of the comments are appended in the body itself for considerations. In conclusions, supported with data might have further improved the standard of the manuscript.Thanks for identifying me as one of the reviewers for the manuscript and giving sufficient time to review.

Reviewer #3: Authors have addressed all the queries in the manuscript and thoroughly improved. The manuscript can be accepted for publication.

7. PLOS authors have the option to publish the peer review history of their article (what does this mean?). If published, this will include your full peer review and any attached files.

Reviewer #1: No

Reviewer #2: No

Reviewer #3: **Yes: **B. Lal

---

## [Author Response · Author response to Decision Letter 1]

23 Nov 2022

Thank you for your useful comments and suggestions on the structure of our manuscript. We are very thankful to the reviewer for thoroughly reviewing our manuscript. These comments and suggestions were helpful in improving our manuscript and we have addressed them in the revised version and highlighted with red color. All the editorial comments have been incorporated throughout the manuscript text as pointed out. Checked all grammatically error, spelling of heading and subheading key words of the manuscript. Retracted references are deleted and revised in both the references list and the manuscript text.

---

## [Editor Report · Decision Letter 2]

7 Dec 2022

Impact of sequential herbicides application on crop productivity, weed and nutrient dynamics in soybean under conservation agriculture in Vertisols of Central India

PONE-D-22-19003R2

Dear Dr. Meena,

We’re pleased to inform you that your manuscript has been judged scientifically suitable for publication and will be formally accepted for publication once it meets all outstanding technical requirements.

Kind regards,

Mohan Lal Dotaniya, Ph.D.

Academic Editor

PLOS ONE
---

## [Editor Report · Acceptance letter]

5 Jan 2023

PONE-D-22-19003R2 

Impact of sequential herbicides application on crop productivity, weed and nutrient dynamics in soybean under conservation agriculture in Vertisols of Central India 

Dear Dr. Meena:

I'm pleased to inform you that your manuscript has been deemed suitable for publication in PLOS ONE. Congratulations! Your manuscript is now with our production department. 

Kind regards, 

on behalf of

Dr. Mohan Lal Dotaniya 

Academic Editor

PLOS ONE